# *AINR*: Adaptive learning of activations for Implicit Neural Representations

## Abstract

Implicit Neural Representations (INRs) provide a continuous function learning framework for discrete signal representations. Using positional embeddings and / or specialized activation functions, INRs have overcome many limitations of traditional discrete representations. However, existing work primarily focuses on the use of a single activation function throughout the network, which often requires an exhaustive search for optimal activation parameters tailored to each signal and INR application. We hypothesize that this approach may restrict the representation power and generalization capabilities of INRs; limiting their broader applicability. In this paper, we introduce AINR, a method that adaptively learns the most suitable activation functions for INRs from a predefined dictionary. This dictionary includes activation functions such as Raised Cosines (RC), Root Raised Cosines (RRC), Prolate Spheroidal Wave Function (PSWF), Sinc, Gabor Wavelet, Gaussian, and Sinusoidal. Our method identifies the activation atom that is mostly matched for each layer of the INR based on the given signal. Experimental results demonstrate that *AINR* not only significantly improves INR performance across various tasks, such as image representation, image inpainting, 3D shape representation, novel view synthesis, super resolution, and reliable edge detection, but also eliminates the need for the previously required exhaustive search for activation parameters, which had to be conducted even before INR training could begin.

## 1 Introduction

Implicit Neural Representations (INRs), also known as coordinate-based neural networks, operate by learning a continuous (implicit) functional representation when provided with the coordinates of an explicit signal representation. In general, an INR is structured as a multilayer perceptron (MLP) with several fully connected layers, where the explicit signal's coordinates serve as the input. Through the learning process of the MLP, the explicit representation is encoded into the weights and biases of the neural network. A distinctive feature of INRs is their versatility in handling different types of signals, from two-dimensional images through three-dimensional shapes and beyond. For example, in the context of images, an INR utilizes the coordinates from a two-dimensional grid to produce the corresponding color values at those coordinates, effectively learning a continuous representation for the image.

INRs stand in contrast to traditional discrete signal representation techniques, offering a more flexible and potentially more efficient means of representing complex signals (Dupont et al., 2021). Once the conversion of an explicit signal representation to an implicit representation through an INR is completed, a continuous functional relationship between the signal's coordinates and its values is established. This learned continuous implicit functional relationship, facilitated by INRs, serves as a robust representation mechanism for the underlying signal, allowing it to perform operations like precise querying of the learned representation and differentiation, etc. In contrast, discrete representations of signals encounter limitations in operations such as querying, which are constrained by quantized interpolations, and also differentiation may not yield desired outputs due to the discrete nature. Therefore, the inherent capabilities of INRs offer significant advantages in accurately representing and manipulating signals compared to discrete representations. In addition, while the memory requirement for conventional representations increases exponentially with the signal resolution, INRs are not tied to the resolution, making this approach highly memory-efficient (Dupont et al., 2021; Strümpler et al., 2022).

Despite the potential advantages of using INRs for the applications mentioned above, their performance critically depends on the architecture of the MLP, particularly the choice of activation function. Traditional activation functions, such as ReLU, Sigmoid, and Tanh, which are commonly used in deep learning models, have shown very poor performance in INRs (Sitzmann et al., 2020; Tancik et al., 2020). This inefficiency is primarily due to their inability to effectively pass the high-frequency components of signals through the network (Yüce et al., 2022). As a solution, Tancik et al. (2020) proposed a fixed coordinate transformation prior to training, commonly referred to as positional embedding, which embeds high-frequency content into the input coordinates of an INR. While positional embeddings can enhance representation, Sitzmann et al. (2020) found that they suffer from limited representational capacity and struggle to generalize effectively. To mitigate these issues, they introduced sinusoidal activations, with a specific frequency and a carefully designed MLP weight initialization, bypassing the need for positional embeddings. Nevertheless, the reliance on exact weight initialization and frequency tuning for sinusoidal activations presents a significant limitation, despite their generalization strength. Ramasinghe & Lucey (2022) relaxed the stringent weight initialization requirements, and further improvement was achieved by using Gabor wavelets as activation functions (Saragadam et al., 2023), which leverages their strong space-frequency localization. Nonetheless, these non-linear activations still require specific activation function parameters to be determined for each discrete signal and INR application, often necessitating an exhaustive grid search (Saragadam et al., 2023). This dependence on pre-selected activation function parameters limits the flexibility of INRs, as these parameters must be known in advance for efficient explicit-to-implicit conversion. Moreover, to the best of our knowledge, prior research has focused only on using a single activation function to improve INR capabilities. This leads to the following questions: Could using multiple activation functions adaptively enhance both the expressiveness and generalization of INRs?, 2). How can we eliminate the prolonged and time-consuming exhaustive grid search process faced by the INR community to determine activation parameters even before training any INR?

To address these existing issues within INRs, we present "Adaptive Learning of Activations for Implicit Neural Representations" (*AINR*), a novel approach that enables INRs to dynamically adjust the activation function for each layer, optimizing it for the given signal. For the adaptive learning process, we propose a dictionary of activation atoms, paired with a matching pursuit-based mechanism (Mallat & Zhang, 1993), which selects the activation atom that is most matched for each layer of the INR. The dictionary includes four new activation "atoms"—Raised Cosines (Alagha & Kabal, 1999), Root Raised Cosines (Joost, 2010), Prolate Spheroidal Wave Functions (Slepian & Pollak, 1961; Landau & Pollak, 1961; 1962; Slepian, 1964; 1978), and Sinc functions (Shannon, 1948)—chosen for their strong space-frequency localization, a feature commonly leveraged in signal and image processing. In addition, we incorporate three widely used activations from the INR literature: Sinusoids (Sitzmann et al., 2020), Gabor Wavelets (Fathony et al., 2020; Saragadam et al., 2023), and Gaussians (Ramasinghe & Lucey, 2022).

To demonstrate the performance of the proposed *AINR*, we present several applications, including image representation, image inpainting, super-resolution, occupancy field representation, novel view synthesis, edge detection, and high-frequency encoding capabilities. Our thorough evaluation of *AINR* shows that it surpasses state-of-the-art INR solutions by a clear margin. Furthermore, our comprehensive ablations reveal that *AINR* eliminates the need for specific activation function parameter determination, previously deemed necessary prior to training any INR. In summary, *AINR* emerges as the new benchmark in the INR field.

## 2 RELATED WORKS

**Activation Functions.** Neural network activation functions, also referred to as transfer functions (Apicella et al., 2021), determine the output of each neuron based on the weighted sum of the inputs they receive from the previous layer. These functions are typically non-linear and aid neural networks in capturing non-trivial functional relationships with a reduced number of nodes (Szandała, 2021). Unlike network's weights and biases, which are updated based on training data, activation functions are typically chosen beforehand and remain unchanged throughout the training process (Lederer, 2021). However, data-dependent activations, i.e., trainable activations, were recently proposed using the classical sigmoid function (Apicella et al., 2021). Since then several other trainable activations have been proposed (Yuen et al., 2021; Dubey et al., 2022). Several

studies have also explored the connections between deep neural networks and activation functions from a frequency perspective (Xu et al., 2019; Benbarka et al., 2022), providing additional insight to understand their behavior and impact on neural network dynamics.

**Compactly Supported and Band-limited Signals** have the property of having non-zero values only within a bounded set of the considered space or in a transformed domain like Fourier. This property is typically desirable in signal processing, communications, and other fields as it comes with efficient approximation, transmission, and recovery properties (Proakis, 2008). Since it is mathematically impossible to have both compact support and band-limitedness at the same time, the next best property is to have some form of *space-frequency concentration*, that is, compact support with rapid frequency decay or band-limited with rapid space decay or rapid decay in both domains. With the advances in INRs, it has been demonstrated that when an activation function has good space and frequency concentration, it not only significantly enhances INR performance but also eliminates the need for specific INR weight initialization (Ramasinghe & Lucey, 2022; Saragadam et al., 2023).

**Implicit Neural Representations (INRs)** have recently garnered attention from the computer vision research community, mainly due to their simplistic network architecture and the performance improvements observed in various vision tasks compared to traditional parameter-heavy vision models (Saragadam et al., 2023; Sitzmann et al., 2020; Cervantes et al., 2022). The emergence of INRs has begun mainly after the introduction of neural radiation fields with ReLU activations(Mildenhall et al., 2021), which has led to multiple follow-up studies(Gao et al., 2022; Molaei et al., 2023), and the use of sinusoidal activation as an alternative to conventional ReLU activations (Sitzmann et al., 2020). Thereafter, Ramasinghe & Lucey (2022) has shown the existence of a broader class of activations that are suitable for INRs. The more recent work, Saragadam et al. (2023) has proposed Gabor Wavelets, which are not only compactly supported but also benefit from exponential damping, as a non-linearity for INRs, and showed improved INR performance compared to previous INRs models.

## 3 METHODOLOGY

### 3.1 FORMULATION OF AN INR

Consider an INR denoted as $F_\theta$, where $\theta$ represents the neural network parameters. $F_\theta$ takes coordinates from a $K$-dimensional space, denoted as $\mathbb{R}^K$, and maps them to a $M$-dimensional signal, denoted as $\mathbb{R}^M$. Therefore, this mapping can be expressed as:

$$F_\theta : \mathbb{R}^K \to \mathbb{R}^M.$$

If $W^{(i)}$ and $b^{(i)}$ are the weight and bias matrices of the $i^{\text{th}}$ layer, the input to the $(i + 1)^{\text{th}}$ layer is given by $\sigma^{(i)}(W^{(i)}x^{(i)} + b^{(i)})$, where $\sigma^{(i)}$ represents the activation function of the $i^{\text{th}}$ layer, and $x^{(i)}$ is the input to the $i^{\text{th}}$ layer. The representation capacity or the learning dynamics of an INR is governed by the activation function $\sigma$ (Sitzmann et al., 2020; Ramasinghe & Lucey, 2022; Saragadam et al., 2023; Tancik et al., 2020). Most studies have used a single activation type for the entire network, i.e., $\sigma^{(i)} = \sigma$ for all $i$. Through extensive experiments, we show that this approach often leads to suboptimal learning outcomes for INRs since constraining the INR to a single activation function throughout the network limits the expressive power of the learned model. Consequently, the model struggles when attempting to generalize to unseen or untrained coordinates, undermining the intended purpose and functionality of INRs. Therefore, this limits the INR's adaptability and effectiveness, whereas robustness and generalization are essential aspects when moving from one representation to another.

### 3.2 DICTIONARY OF ACTIVATIONS FOR *AINR*

We adopt a dictionary comprising of seven activation functions. This includes two functions with rapid decay in both the spatial and Fourier domains (Complex Gabor Wavelets and Gaussian) and five band-limited functions (Sinc, Raised Cosine, Root Raised Cosine, Sinusoid, and PSWF).

1. **Sinc Function:** The sinc function is the Fourier transform of a rectangular pulse in the Fourier domain (in digital communication literature, this is also referred to as the Nyquist

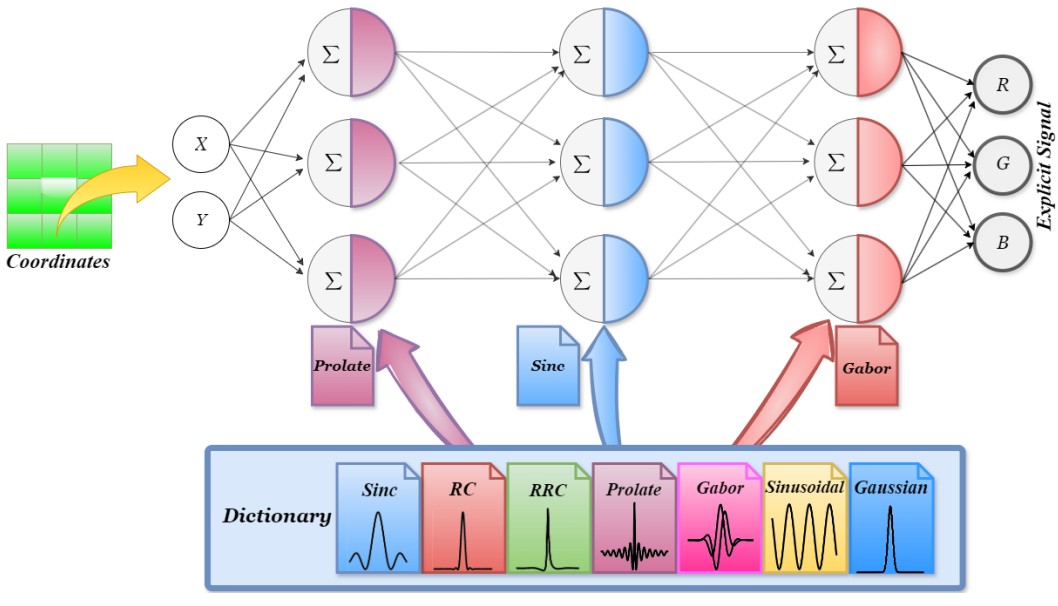

Figure 1: **Illustration of *AINR*.** Existing INRs have been limited to a single activation function type, and this could limit the expressive power and generalizability of INR models. In this study, we introduce a dictionary of activation functions, along with a mechanism to select the most suitable activation from the dictionary for each layer. As each activation sequence is tailored to the given signal, *AINR* emerges as a most effective and generalizable INR.

pulse (Shannon, 1948)). It is defined as, $\text{sinc}(\alpha x) = \frac{\sin(\alpha x)}{\alpha x}$, and it decays as $\frac{1}{\alpha x}$, where $\alpha$ is a parameter.

2. **Raised Cosine (RC):** It is another band-limited function whose decay in the space domain is of order $x^{-2}$, hence faster than the Sinc. Defined by the parameters $\alpha$, $\beta$, and $\gamma$, the raised cosine activation atom takes its form as:

$$\frac{\text{sinc}(\alpha x)\cos(\beta x)}{1 - |\gamma|x^2}.$$

3. **Root Raised Cosine (RRC):** This is a modified version of the raised cosine obtained by taking the square root of the frequency response of the raised cosine pulse. This modification improves the decay of the signal. With $\alpha$, $\beta$, $\gamma$, $a$, and $b$ as the parameters, the root-raised cosine activation atom is defined as,

$$\frac{a\sin(\alpha x) + b\cos(\beta x)}{1 - |\gamma|x^2}.$$

4. **Prolate Spheroidal Wave Function (PSWF):** These are the solutions to the Helmholtz equation in prolate spheroidal coordinates. The Helmholtz equation in prolate spheroidal coordinates can be transformed to the following ordinary differential equation, where $m$, $n$, and $c$ are parameters, and $R_{mn}(c, x)$ are the PSWFs:

$$(x^2 - 1)\frac{d^2 R_{mn}(c, x)}{dx^2} + 2x\frac{dR_{mn}(c, x)}{dx} - \left[\lambda_{mn}(c) - c^2 x^2 + \frac{m^2}{x^2 - 1}\right]R_{mn}(c, x) = 0$$

This differential equation arises in the context of bandlimited signals when the signal that has the highest possible energy concentration within a given interval (Gonzalez, 2018). Although finding a closed-form solution is difficult, a discretized approximation for PSWFs is taken in this study. To define it as an activation atom, the natural cubic spline approximation has been used.

5. **Complex Gabor Wavelet:** Gabor Wavelets involve a Gaussian-modulated cosine or sine wave. They offer rapid decay in both spatial and frequency domains, and have already been

employed for INRs showing better performance compared to sinusoidal activations. With $\alpha, \gamma$ as the parameters, the Gabor wavelet activation atom is defined as $e^{j\alpha x - |\gamma| x^2}$

6. **Gaussian:** Similar to Complex Gabor Wavelet, Gaussian functions also offer rapid decay in both spatial and frequency domains, and have been already utilized in INRs. The Gaussian activation atom is defined as, $e^{-|\gamma| x^2}$ where $\gamma$ is a parameter.

7. **Sinusoid:** Sinusoidal functions are band-limited, and have been used in INRs. The sinusoidal activation atom is defined as $\sin(\alpha x + \beta)$, where $\alpha, \beta$ are parameters.

When implementing AINR, the real part of the complex Gabor Wavelet has been taken. The distinct spatial characteristics of each activation function can be observed from figure 11 in Appendix, where the variation of the activation function value with spatial distance is depicted.

## 3.3 PREMISE OF *AINR*

AINR begins by constructing a dictionary of predefined activation functions, as described in section 3.2, each equipped with trainable parameters. These activation atoms have their parameters randomly initialized, drawn either from a uniform or normal distribution. *AINR* is then initialized as a single-layer MLP, with input and output dimensions customized to match the explicit signal representation. The algorithm iterates through each activation function in the dictionary, applying it as the non-linearity for the hidden layer over a specified number of training epochs. During this process, the algorithm tracks the performance of each activation function by calculating the mean square loss between two representations, saving the parameters of the activation that achieves the lowest loss. After all activation atoms have been evaluated, the algorithm selects the activation function that produces the minimum loss for the first layer.

Once the most suitable activation function for the first-hidden layer is identified based on the minimum loss criterion, it is fixed as the non-linearity for the first-hidden layer, along with its associated parameters. *AINR* then proceeds to add a second-hidden layer. It again starts the MLP training process afresh, with one key difference: the first-hidden layer's activation function and its optimized parameters, which minimized the loss when using a single hidden layer, are retained. The algorithm then tests each activation function from the dictionary as the non-linearity for the second-hidden layer, again over a predetermined number of epochs. Similarly, the performance of each activation is recorded. At the end of this sweep, based on the performance, the activation that gives the minimum loss is selected, and fixed as the second layer's non-linearity. This process continues for all the hidden layers. Figure 1 illustrates the process of selecting an optimal activation sequence for image representation tasks by *AINR* through the activation function dictionary. For a better understanding of the *AINR*'s training process, please refer the pseudo-code provided (See section A.4.1 in Appendix) along with section 3.3.

## 3.4 ACTIVATION FUNCTION PARAMETER INITIALIZATION

The performance of an INR with parametric activation functions is significantly dependent on the initialization of the activation parameters. Inappropriate initialization often leads to poor performance in almost all tasks that involve existing INRs. Previous studies have used extensive grid searches to determine the activation parameters (Saragadam et al., 2023) for each application, but the effectiveness of this method is highly dependent on the diversity of the signal and may perform poorly with signals that differ from those used during parameter determination. In contrast, AINR randomly initializes the activation function parameters, allowing the network to learn the optimal parameters for each application and signal during optimization. This adaptability enables *AINR* to optimize based on the specific characteristics of the signal. Experimental results demonstrate that *AINR* outperforms existing INRs, including WIRE (Saragadam et al., 2023), SIREN (Sitzmann et al., 2020), GAUSS (Ramasinghe & Lucey, 2022), and MFN (Fathony et al., 2020), in both performance and generalization in various tasks.

## 4 EXPERIMENTAL RESULTS

### 4.1 IMAGE REPRESENTATION

As mentioned earlier, a direct application of *AINR* is learning an implicit representation of an image, commonly referred to as image representation. In this framework, the network is provided with normalized coordinates of a signal without any positional embedding, and the*AINR* is trained to predict the corresponding RGB values. First, to clearly illustrate how *AINR* functions, we chose an image that exhibits high spatial variation and has a broad frequency range. This image is shown in the figure on the left of the top row of figure 2. Second, for a more comprehensive evaluation, the representation capacity of *AINR* is evaluated across the Kodak (Kodak) data set. The resulting PSNR for each image, along with the baseline results, is shown in figure 3. For the average PSNR and the decoded representations of each method, refer to section A.5.1 in the Appendix.

As shown in the top row of figure 2, *AINR* achieves the highest PSNR and SSIM values, indicating the lowest distortion and best preservation of structural information, texture, and contrast compared to existing INRs. For this experiment, we used all the activations defined in section 3.2. As detailed in section 3.3, *AINR* begins with a single hidden layer and searches the dictionary to determine which activation produces the highest PSNR (or lowest loss). This process is carried out for 100 epochs for each activation atom in the dictionary. Upon identifying the activation that is mostly matched to the image according to the loss criterion, Sinc in this case, it locks this activation for the first hidden layer (bottom left, figure 2). Subsequently, *AINR* adds the second hidden layer and resumes training the entire network afresh while keeping the first layer's activation and parameters frozen, adjusting only the activation of the second-hidden layer at every 100 epochs. At the end of this phase, it determines the activation that provides the highest PSNR (or lowest loss) for the second hidden layer, which in this instance is Gaussian (bottom middle, figure 2). Following this, *AINR* introduces the third hidden layer and begins training the network afresh now while keeping both first and second layers' activations and their parameters kept frozen, this time modifying the activation of the third layer at every 200 epochs. Upon completion of training, *AINR* identifies the activation for the third hidden layer that results in the highest PSNR (or minimum loss), which, for this stage, is the Gabor Wavelet (Bottom right, figure 2). Therefore, the matched sequence of activation atoms for the Parrot image, as identified by *AINR*, is Sinc, Gaussian, and Gabor wavelet. Note that although *AINR* determines the most suitable activation for each layer based on loss, PSNR plots are used for illustrative purposes.

Considering the bottom row of figure 2, we can conclude that, once the network has the matched sequence of activation functions determined by *AINR*, INRs start showcasing a faster convergence. In the case of *AINR*, when the first two layers' activation functions are determined, it only needs at most 200 epochs to obtain a minimum loss between implicit and explicit representations. Therefore, showing a much faster convergence rate compared to the current state-of-the-art INRs. As can be evidenced from both figure 2 and figure 3 , an INR achieves the highest accuracy metrics when its activation functions are customized for a specific signal, rather than using a pre-optimized, uniform activation sequence like in WIRE or SIREN throughout the INR. This thorough evaluation confirms our hypothesis that tailoring the activation function sequence to the signal significantly enhances the INR's performance, even when the activation parameters are randomly initialized.

### 4.2 IMAGE INPAINTING

Unlike explicit discretized signal representations, an INR learns a continuous implicit representation of a given signal through the MLP training process. Therefore, once the corresponding explicit representation is encoded into the weights and biases of an INR, one should be able to query the model as desired. The inpainting task serves as a good measure of INRs to assess whether the model is overfitted, as the primary purpose of adopting a new representation is to generalize it through learned continuous mapping. To demonstrate the functionality of *AINR* for inpainting, we selected an image with intricate details, shown in the top left of figure 5. The adjacent image on the right shows the same image with a text mask applied. Additionally, we evaluated *AINR*'s inpainting performance on the Kodak dataset to provide a more comprehensive assessment. The PSNR results for each image, along with baseline comparisons, are shown in figure 4. For the inpainted images, and average performance metrics on image inpainting on the Kodak dataset please refer section A.5.2 in Appendix. For this experiment, the newly introduced activations i.e., RC, RRC, Sinc, PSWF, and

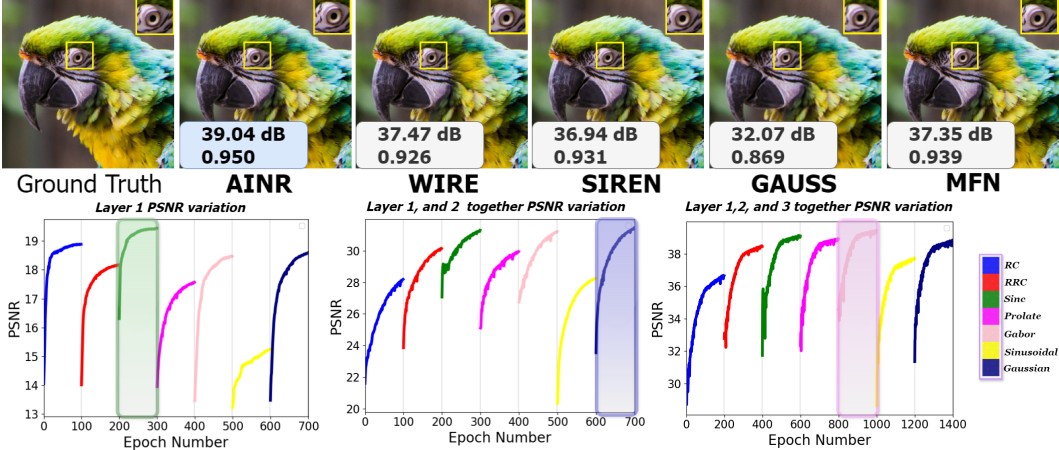

Figure 2: **Image representation capacity of *AINR***: The top row depicts the image reconstruction using various types of INRs. *AINR* stands out as the INR that achieves the highest PSNR and SSIM metrics, indicating minimal distortion and maximum preservation of structural information. The bottom row illustrates how *AINR* achieves these results through sequential training. By tailoring activations to the specific image, the corresponding sequence is Sinc, Gaussian, and Gabor Wavelet under randomly initialized activation parameters.

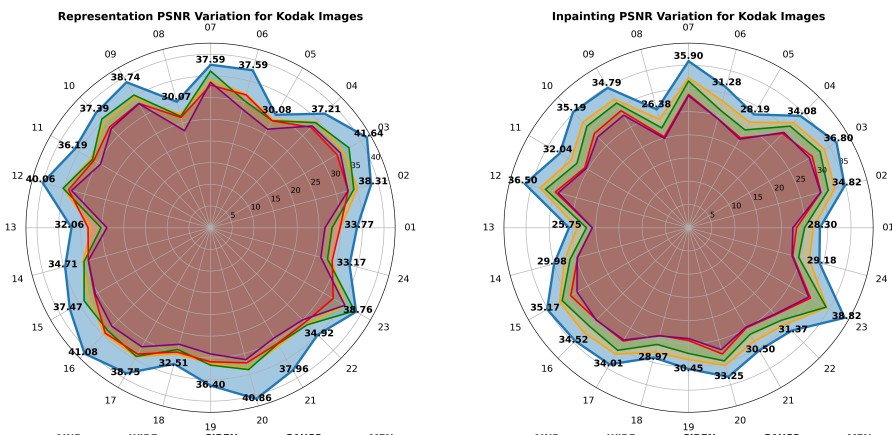

Figure 3: **Image representation capabilities of *AINR* on the Kodak dataset**

Figure 4: **Image inpainting capabilities of *AINR* on the Kodak dataset**

Gabor Wavelet have been used to showcase the effectiveness of these activations. The bottom row of figure 5 showcases the PSNR performance observed in each layer when following the procedure in section 3.3. It should be noted that in the case of image inpainting, the loss calculation for deciding the activation is based on the partial image data. The results clearly demonstrate that AINR delivers the cleanest and most visually coherent image inpainting outcomes compared to all existing INRs. Beyond producing the most visually coherent images, *AINR* also achieves the highest PSNR and SSIM values for the inpainting tasks.

### 4.3 OCCUPANCY FIELDS REPRESENTATION

As INRs offer a continuous functional mapping from low-dimensional coordinate space to signal space, they can be used to effectively represent three-dimensional signed distance fields. In this scenario, the mapping extends from the three-dimensional space to a one-dimensional space, where the signal space is represented by binary values: either 1 or 0. Here, 1 denotes that the signal lies within the specified region, while 0 indicates its absence in the given region. For this experiment, two datasets, Thai Statue and Stanford Lucy, were obtained from Stanford 3D datasets (Stanford University Computer Graphics Laboratory). The sampling procedure followed the method described

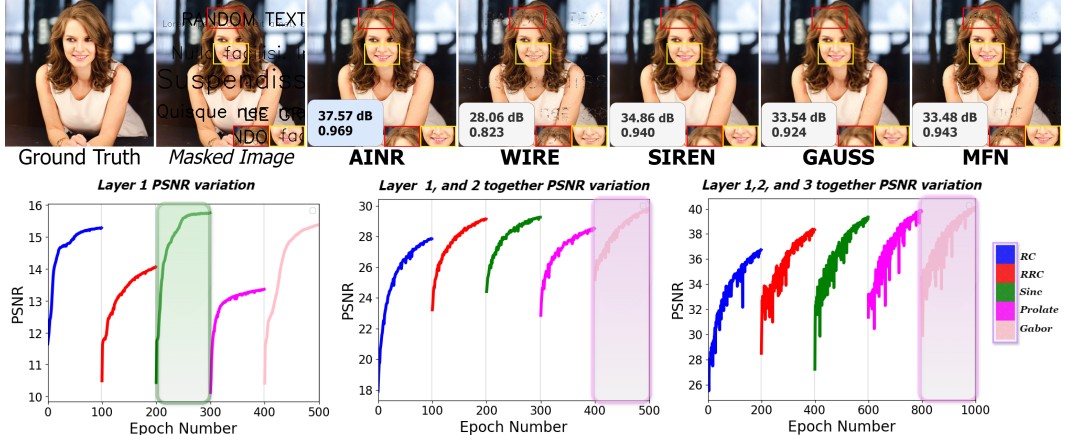

**Figure 5: Image inpainting capabilities of *AINR*.** The top row displays the recovered images using various types of INRs. AINR stands out as the INR that not only achieves the highest PSNR and SSIM metrics but also yields the most visually coherent inpainting outcome. The higher metrics indicate that *AINR* has restored the image with minimal distortion and maximum preservation of structural information. The bottom row illustrates how *AINR* achieves this result through sequential training. By tailoring activations to the specific image, the corresponding sequence consists of Sinc, Gabor Wavelet, and Gabor Wavelet under randomly initialized activation parameters.

in Saragadam et al. (2023), using a $512 \times 512 \times 512$ grid. Voxels inside the volume were assigned a value of 1, while those outside the volume were assigned a value of 0. The sampled volumes for Stanford Lucy and the Thai Statue are displayed in the first column of the 1st and 2nd rows, respectively, in figure 17.

Figure 6 shows the decoded representations for each INR along with the ground truth. As can be clearly seen *AINR* achieves the highest Intersection over Union (IoU) metric, demonstrating the greatest representation capacity among all existing INRs regardless of the occupancy field. A closer examination of decoded statues reveal that *AINR* precisely encodes intricate high-frequency details. In contrast, INRs like WIRE[1] and SIREN tend to converge toward a low-pass representation, high-lighting the challenge of encoding rapidly varying, detailed features in these models. These findings clearly indicate that not only for images but for any signal, when the matched sequence of activations is identified, an INR can accurately learn the implicit representation. In these experiments, *AINR* determined the matched sequence of activations for Stanford Lucy as Sinc, RRC, and PSWFs for the first, second, and third layers, respectively. For the Thai Statue, the activations were RC, RRC, and Gabor Wavelet for the respective layers. The complete occupancy fields corresponding to figure 6 is shown in section A.5.3 in the Appendix.

## 4.4 NEURAL RADIANCE FIELDS

INRs have gained popularity in the computer vision community, largely due to the impact of NeRFs (Mildenhall et al., 2021). In which, a 3 dimensional scene is encoded in an INR by inputting the viewer's spatial coordinates $(x, y, z)$ and viewing angles $(\theta, \phi)$ into the network with the aid of collection of images captured around the scene. The INR is tasked with predicting the color and density at those locations. When the INR is trained, the INR can generate unseen perspectives from new spatial positions and viewing angles which are not present in the training data. For this experiment, we utilized a vanilla NeRF architecture with Chair, and Hotdog datasets. Each dataset has 100 training, and 200 testing images. Once the network is trained, the testing PSNRs across the testing views are averaged. The top and bottom rows of figure 7 show novel views generated from the trained INR models on the Chair and Hotdog datasets, respectively. Additional novel views are provided in section A.5.4 in the Appendix.

---

[1]∗ Reproduced result with 300 hidden neurons

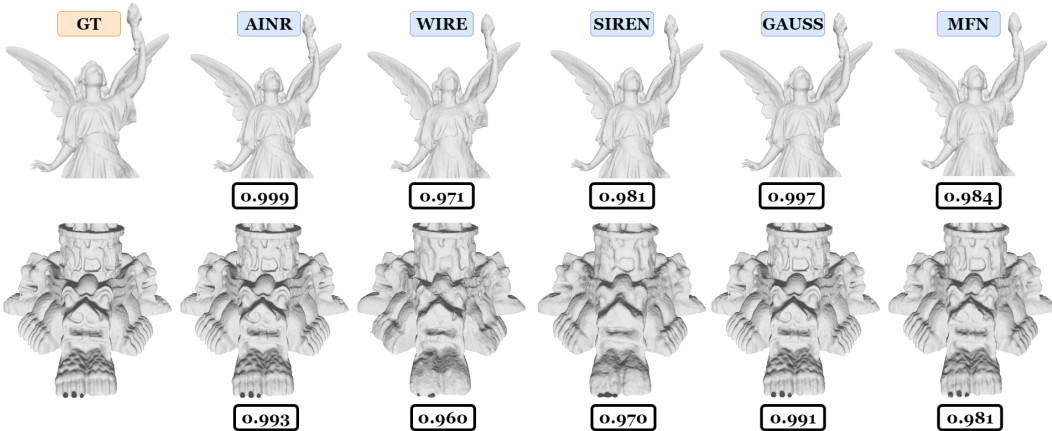

Figure 6: **Occupancy fields representation capacity of AINR**: The image illustrates the reconstruction capabilities of various INRs for occupancy volumes. AINR stands out as the INR that not only achieves the highest IoU metric but also the INR which preserves the highest amount of fine details in its weights and biases. Unlike other INRs, AINR does not tend toward low-pass representations as it encodes signals by tailoring a sequence of activations for the given signal.

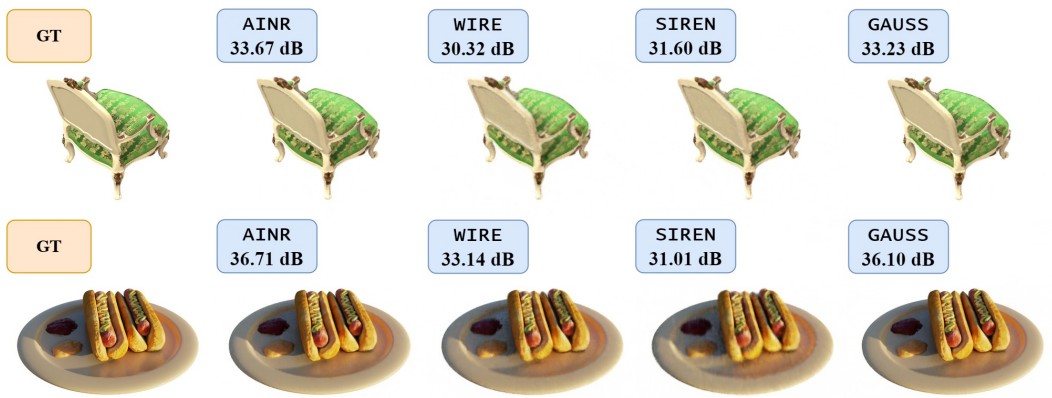

Figure 7: *AINR's novel view synthesis capabilities*: *AINR* consistently achieves the highest performance metrics and captures more intricate details than the baselines. For the chair dataset, *AINR* accurately produced the fine textures, carvings, and lighting effects, closely matching the ground truth. Similarly, for the hotdog dataset, *AINR* preserves the texture, shadows, and reflections with greater fidelity, delivering sharper and more realistic results than WIRE, SIREN, and GAUSS, which tend to over-smooth these details.

## 4.5 EFFECT OF ACTIVATION PARAMETER INITIALIZATION

As outlined in section 3.4, the performance of conventional INRs heavily depends on the initialization of activation function parameters. In contrast, AINR identifies the most matched activation sequence for a specific task without needing precise initialization. To substantiate this claim, we have sourced activation function parameters from both uniform and normal distributions. The primary reason for selecting these distributions is to understand how INRs perform when parameters are derived from distributions that are either spread evenly across a range or centered around a mean value. A uniform distribution over $[a, b]$ is denoted as $U(a, b)$, and a normal distribution with mean $\mu$ and standard deviation $\sigma$ as $\mathcal{N}(\mu, \sigma)$. The results in table 1 show the average PSNR (in dB) from five trials on the Parrot image in figure 2, with variability expressed as the standard deviation next to the $\pm$ symbol. The bold number indicates the highest PSNR, and the following number represents the lowest standard deviation.

Table 1: The PSNR variation of existing INRs when activation functions are drawn from different probability distributions.

| Distribution | AINR | WIRE | SIREN | GAUSS |
|---|---|---|---|---|
| $U(0,1)$ | **39.63 $\pm$ 0.53** | 21.11 $\pm$ 1.01 | 17.48 $\pm$ 2.54 | 16.92 $\pm$ 3.16 |
| $U(-1,1)$ | **39.51 $\pm$ 0.67** | 20.74 $\pm$ 1.51 | 15.66 $\pm$ 2.61 | 18.58 $\pm$ **0.41** |
| $U(-10,10)$ | **40.07 $\pm$ 0.66** | 37.35 $\pm$ 1.02 | 23.96 $\pm$ 6.81 | 24.01 $\pm$ 8.45 |
| $U(-100,100)$ | **36.32 $\pm$ 1.83** | 24.05 $\pm$ 6.80 | 35.70 $\pm$ 2.84 | 22.45 $\pm$ 1.96 |
| $\mathcal{N}(0,1)$ | **39.97 $\pm$ 0.78** | 22.41 $\pm$ 4.03 | 15.93 $\pm$ 2.58 | 17.06 $\pm$ 3.17 |
| $\mathcal{N}(0,10)$ | **38.92 $\pm$ 1.26** | 32.86 $\pm$ 4.79 | 28.87 $\pm$ 4.14 | 27.46 $\pm$ 4.78 |

As illustrated in table 1, AINR emerges as the only INR which is capable of delivering consistent PSNR across various distributions while exhibiting minimal variation around the mean. AINR not only maintains PSNR consistency but also records the highest PSNR values. In contrast, WIRE demonstrates commendable performance exclusively under the $U(-10, 10)$ distribution, suggesting its activation parameters require initialization within a narrow range (-10 to 10) for the tested parrot image. Similarly, SIREN shows enhanced performance when its activation parameters are selected from $U(-100, 100)$ distribution. These observations underline the dependency of INRs like WIRE, SIREN, and GAUSS on specific initial conditions for their activation parameters to guide the network towards convergence.

### 4.6 ADDITIONAL EXPERIMENTS AND ABLATION STUDIES

Comprehensive experiments on image super-resolution, edge detection, and high-frequency encoding are presented in the Appendix, along with details of the experimental setup and ablation studies that evaluate *AINR*'s performance in relation to hidden neurons, layers, learning rates, weight initialization, and positional encoding. Additionally, we provide explanations of training curves, variations in activations within the spatial domain, and how *AINR* differs from baseline methods. Further results on image representation, inpainting, occupancy fields, and novel view synthesis from multiple viewpoints are also included.

## 5 CONCLUSION

Existing INR methodologies are often constrained by the use of a single activation function throughout the neural network, limiting their expressive power and generalizability. Furthermore, current INRs require prior knowledge of activation function parameters, which are typically determined through grid searches. However, these parameters can be suboptimal when the INRs encounter signals with characteristics that differ from those used during parameter selection. In this work, we introduce a dictionary of activation functions that encompasses seven nonlinearities, including four that have not previously been utilized in INRs: Raised Cosine, Root Raised Cosine, Prolate Spheroidal Wave Function, and Sinc function. The other three activations, i.e., Gabor Wavelet, Gaussian, and Sinusoid, are well-known in the INR field. Along with the activation dictionary, we proposed a non-exhaustive mechanism based on the matching-pursuit algorithm to automatically identify the matched sequence of activations for any given INR task. Our extensive numerical experiments demonstrate that the proposed method, *AINR*, achieves convergence even with random initialization of activation function parameters, in contrast to existing INRs that typically require prior knowledge or a search for optimal parameters. Additionally, *AINR* demonstrates superior representation and generalization capabilities by adaptively selecting the activation sequence that minimizes the loss between implicit and explicit representations. This adaptability allows *AINR* to outperform current INRs, establishing it as a new state-of-the-art in the field.

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
