# OpenReview forum: "AINR: Adaptive Learning of Activations for Implicit Neural Representations"
_ICLR.cc/2025/Conference — ICLR 2025 Conference Withdrawn Submission_

### Official Review · Reviewer_rQSg · 2024-10-30

**Soundness:** 2
**Presentation:** 3
**Contribution:** 2
**Rating:** 3
**Confidence:** 4

**Summary:**

The authors introduce AINR, a method that adaptively learns the most suitable activation functions for INRs from a predefined dictionary. This dictionary includes 7 activation functions. Experimental results demonstrate that AINR not only significantly improves INR performance, but also eliminates the need for the previously required exhaustive search for activation parameters.

**Strengths:**

1. The method is highly innovative, proposing the use of different activation functions across various layers in the MLP to enhance performance.

2. 7 distinct activation functions were considered.

3. 6 different hyperparameter random distribution functions were evaluated.

4. The method demonstrated a strong performance advantage in the experiments presented in the paper.

5. Numerous related experiments and explanations were provided in the supplementary materials.

**Weaknesses:**

1. The potential issues of mixed activation functions should be addressed. Serial use of activations like SINE adds complexity in initialization, and altered data distributions at any layer may require adjusted initialization parameters.

2. The impact of random seeds should be considered, as they can affect activation functions and hyperparameters, influencing algorithm performance.

3. Fitting time should be a performance metric. With 7 activations, assuming 10 hyperparameter options each and 5 repetitions, the proposed algorithm’s runtime could be 350 times that of the comparison.

4. If fitting time isn’t included, fairness suggests giving comparison algorithms more time up to 350 times. Would performance advantages remain significant?

5. Each layer’s optimal activation function and hyperparameters should adapt to different convergence times instead of stopping at 100 epochs uniformly. According to Figure 5 in the manuscript, different layers and activation functions exhibit considerable variation in convergence times.

6. The greedy algorithm’s ability to find a global optimum should be evaluated against other decision methods.

7. Comparison with algorithms using alternative hyperparameters should be considered.

**Questions:**

1.  Could you please discuss potential challenges or mitigations for using mixed activations, or to provide empirical evidence that these issues don't significantly impact performance in practice.

2. Could you please report training times and discuss the tradeoffs between performance gains and increased computational cost. This would help readers better understand the practical implications of using AINR.

3. For Weakness 4, could you please either conduct this experiment or explain why it might not be necessary or feasible.

4. Could you please explore adaptive stopping criteria for each layer/activation, or explain why a fixed number of epochs was chosen despite the potential for suboptimal convergence.

---

### Official Review · Reviewer_7jB8 · 2024-11-02

**Soundness:** 3
**Presentation:** 3
**Contribution:** 4
**Rating:** 6
**Confidence:** 5

**Summary:**

This paper addresses the challenge of selecting an optimal activation function for implicit neural representation, which significantly impacts model performance and spectral bias. Rather than applying a single activation function throughout the network, the authors propose using a dictionary of activation functions—each with unique time- and frequency-domain behaviors—and selecting the most effective one for each layer based on performance. This approach also reduces the reliance on weight initialization, as demonstrated by experimental results. The effectiveness of the proposed method is validated through a variety of experiments.

**Strengths:**

1. The paper provides various experiments.
2. Good writing: The paper is well-written and easy to follow.
3. Excellent performance in selected cases.

**Weaknesses:**

The paper lacks enough mathematical support, including the following points:
1. Assuming weight initialization is a significant concern, your approach only considers two distributions (uniform and normal) to support your claim. However, the space of possible distributions is infinite-dimensional. A possible better idea may be exploring Gaussian Mixture Models and searching over their coefficients. Additionally, your demonstration is primarily qualitative. It would be beneficial to include a statistical test such as hypothesis testing, ANOVA, Kruskal-Wallis, or Mann-Whitney U for quantitative rigor.
2. The incremental algorithm proposed in Section 3.3 (premise of AINR), lacks a solid mathematical foundation. There is no theoretical basis to assume that a neural network trained with $n$ layers will function effectively with $n+1$ layers. Consequently, your method appears largely heuristic.

**Questions:**

1. The authors have claimed that their model removes the need for weight initialization. However, in most previous works (such as SIREN), initialization follows a specific mechanism (having some mathematical support). Thus, previous models didn’t face substantial issues with initialization, aside from minor computational concerns, which are generally negligible since initialization occurs only once before training. I understand that it’s an advantage, but I kindly request you to explain it more.
2. Including an additional graph showing PSNR and SSIM for different methods across different epochs would be useful, as some methods may outperform others at higher epochs.
3. I wonder how your method would compare to hash-encoding [1], given its strong performance across all INR-related tasks.

[1] Instant Neural Graphics Primitives with a Multiresolution Hash Encoding

---

### Official Review · Reviewer_8B2b · 2024-11-03

**Soundness:** 1
**Presentation:** 2
**Contribution:** 2
**Rating:** 3
**Confidence:** 5

**Summary:**

Aiming at improving the performance of implicit neural representations, the authors proposed an algorithm to iteratively select (from a pre-selected dictionary of activation functions) the best-performing activation function in each layer, starting from the first layer and proceed layer by layer. The resulting INR performs slightly better than baseline SIREN, GAUSS and WIRE-based INRs. The authors also attached a brief discussion of the impact of parameter initialization on the results.

**Strengths:**

The paper host two major strengths:
1. From the results reported by the authors, the performance of the proposed INR surpasses the baseline INRs.
2. The approach can be easily expanded to cover more activation functions as candidates for selection.

**Weaknesses:**

Quite a series of weaknesses can be observed, attached in the order of importance below:
1. The baselines are either old or not strong. SIREN is from 2020, GAUSS is from 2022, and whilst WIRE is from 2023, it has been reported in multiple recent works (plus in this one also) that it is not a strong baseline, and sometimes even performing worse than the older SIREN and GAUSS. It is important that the authors compare with more recent, stronger baselines, for example FINER [1] or sinc [2]. In particular the sinc-based INR [2] is well worth comparing with since it is within the dictionary that the authors draw activation function from.

[1] Liu, Zhen, et al. "FINER: Flexible spectral-bias tuning in Implicit NEural Representation by Variable-periodic Activation Functions." Proceedings of the IEEE/CVF Conference on Computer Vision and Pattern Recognition. 2024.

[2] Saratchandran, Hemanth, et al. "A sampling theory perspective on activations for implicit neural representations." Forty-first International Conference on Machine Learning. 2024

2. The authors did not report training and inference speed. It is rather important to report the speed since one could almost always improve in metrics by increasing the complexity of the activation function [3]. If the improvement in performance is at the cost of training and inference speed (which is probably the case for this work at least for the training part), then such improvement is questionable.

[3] Liang, Senwei, et al. "Reproducing activation function for deep learning." Communications in Mathematical Sciences 22.2 (2024): 285-314.

3. The idea of identifying the best-performing activation function layer-by-layer can be questionable. Note that the initialization of SIREN, for example, is different between the first layer and the other layers, where the first layer serves to adjust the distribution of the activation such that starting from the second layer the distribution of the activations are stable over each layer. As a result, a SIREN-based INR with less than two hidden layers will not achieve the designed performance, and therefore it is questionable to select or exclude SIREN in the sweep to decide the activation function for the first layer.

4. The author do provide a brief discussion of initialization on the results but not adequate. Note that different activation functions require very different initialization and have very different assumptions on the distribution of the incoming signal from the previous layer. It is quite important to make a careful discussion with these in consideration.

**Questions:**

See weaknesses.

---

### Official Review · Reviewer_7Y53 · 2024-11-04

**Soundness:** 2
**Presentation:** 3
**Contribution:** 2
**Rating:** 3
**Confidence:** 5

**Summary:**

The paper introduces AINR (Adaptive Learning of Activations for Implicit Neural Representations), an innovative framework that adaptively selects suitable activation functions from a predefined dictionary for each layer of the network. This new approach aims to address the limitations of existing methods that use a single fixed activation function throughout the network, which often require exhaustive hyperparameter tuning.

**Strengths:**

- The paper is very clearly written and presents its contributions in a highly succinct way.

- The paper validates its approach on diverse tasks to verify the robustness of the proposed method.

**Weaknesses:**

- The paper generally compares its method with earlier methods, it lacks comparisons to some recent methods (e.g. FINER [1], Incode [2], SL2A-INR [3]) which propose new activation (and in some cases adaptive activation) functions for INRs.

- Another weakness of this paper is that they do not show if the proposed method has less spectral bias than SIREN and other methods, which is the core deficiency of INRs.

- I was hoping to provide more theoretical or experimental analysis on the spectral bias, such as using Neural Tangent Kernel tools for their approach.

- How does your method compare with others in terms of complexity? Does it add any complexities to the network?


[1]Liu Z, Zhu H, Zhang Q, Fu J, Deng W, Ma Z, Guo Y, Cao X. FINER: Flexible spectral-bias tuning in Implicit NEural Representation by Variable-periodic Activation Functions. InProceedings of the IEEE/CVF Conference on Computer Vision and Pattern Recognition 2024 (pp. 2713-2722).

[2]Kazerouni A, Azad R, Hosseini A, Merhof D, Bagci U. INCODE: Implicit Neural Conditioning with Prior Knowledge Embeddings. InProceedings of the IEEE/CVF Winter Conference on Applications of Computer Vision 2024 (pp. 1298-1307).

[3]Heidari M, Rezaeian R, Azad R, Merhof D, Soltanian-Zadeh H, Hacihaliloglu I. Single-Layer Learnable Activation for Implicit Neural Representation (SL $^{2} $ A-INR). arXiv preprint arXiv:2409.10836. 2024 Sep 17.

**Questions:**

Please refer to the weaknesses section.

---

### Note · Authors · 2024-11-15

I have read and agree with the venue's withdrawal policy on behalf of myself and my co-authors.